# Role of the Immune System Elements in Pulmonary Arterial Hypertension

**DOI:** 10.3390/jcm10163757

**Published:** 2021-08-23

**Authors:** Michał Tomaszewski, Dominika Bębnowska, Rafał Hrynkiewicz, Jakub Dworzyński, Paulina Niedźwiedzka-Rystwej, Grzegorz Kopeć, Ewelina Grywalska

**Affiliations:** 1Department of Cardiology, Medical University of Lublin, 20-954 Lublin, Poland; tomaszewskimd@gmail.com; 2Institute of Biology, University of Szczecin, 71-412 Szczecin, Poland; dominika.bebnowska@usz.edu.pl (D.B.); rafal.hrynkiewicz@usz.edu.pl (R.H.); 3Department of Inorganic Chemical Technology and Environment Engineering, West Pomeranian University of Technology in Szczecin, 70-322 Szczecin, Poland; jakub.dworzynski7@gmail.com; 4Pulmonary Circulation Center, Department of Cardiac and Vascular Diseases, Faculty of Medicine, Jagiellonian University Medical College, 31-008 Kraków, Poland; grzegorzkrakow1@gmail.com; 5Centre for Rare Cardiovascular Diseases, Department of Cardiac and Vascular Diseases, John Paul II Hospital, 31-202 Krakow, Poland; 6Department of Clinical Immunology and Immunotherapy, Medical University of Lublin, 20-093 Lublin, Poland; ewelina.grywalska@gmail.com

**Keywords:** cytokines, immune checkpoints, immune disorders, lymphocytes, pulmonary arterial hypertension

## Abstract

Pulmonary arterial hypertension (PAH) is a relatively rare disease, but, today, its incidence tends to increase. The severe course of the disease and poor patient survival rate make PAH a major diagnostic and therapeutic challenge. For this reason, a thorough understanding of the pathogenesis of the disease is essential to facilitate the development of more effective therapeutic targets. Research shows that the development of PAH is characterized by a number of abnormalities within the immune system that greatly affect the progression of the disease. In this review, we present key data on the regulated function of immune cells, released cytokines and immunoregulatory molecules in the development of PAH, to help improve diagnosis and targeted immunotherapy.

## 1. Introduction

Pulmonary hypertension is a condition where the pressure in the pulmonary bed abnormally increases due to pulmonary vascular disease, lung parenchyma and heart disease. The diagnostic criterion is ≥20 mmHg mean pulmonary artery pressure (mPAP) recorded by means of direct, invasive hemodynamic measurements [1]. This condition may be a manifestation of many clinical situations, e.g., it is associated with the aggravation of the underlying disease (left ventricular failure, acquired heart defects and respiratory system diseases) to which the therapy was administered. In other cases (HIV infection, certain drugs and toxins, gene mutations and systemic diseases of connective tissue), we can only talk about factors in the development of arterial hypertension, as its severity is related to the advancement of changes in the pulmonary bed [2]. These cases are collectively known as “pulmonary arterial hypertension”. This presents the greatest therapeutic challenge, as changes in the pulmonary arterioles lead to increased pulmonary resistance and right ventricular failure [2,3].

The incidence of PAH varies between 2–7.6 cases per million adults per year, and the incidence ranges between 11–26 cases per million adults [2]. The incidence is four times higher in women than in men, although the survival rate is paradoxically worse in men than in women [2,3,4,5,6]. An updated clinical classification of pulmonary hypertension was presented during the 6th World Congress on Pulmonary Hypertension in 2018. Therein, five basic etiological groups of pulmonary hypertension were distinguished: “pulmonary arterial hypertension”, “pulmonary hypertension caused by left heart disease”, “pulmonary hypertension due to pulmonary diseases and/or hypoxia”, “chronic thromboembolic pulmonary hypertension and other pulmonary artery stenosis” and “unexplained pulmonary hypertension and/or multifactorial pathomechanisms” [6,7,8]. The diagnosis is based on the results of invasive hemodynamic measurements: mPAP ≥ 20 mmHg, PCWP ≤ 15 mmHg and PVR > 3 mmHg [1].

Pathological changes in the vessels of the pulmonary circulation, regardless of their causes, lead to enhanced pulmonary resistance, which results in an increase in the afterload of the right ventricle, its enlargement and hypertrophy. As a consequence, functional tricuspid valve insufficiency develops, which, in the end stage of the disease, leads to the low relapse syndrome and the patient’s death [3].

The current treatment options include PDE-5 inhibitors, soluble guanylate cyclase stimulators, endothelin receptor antagonists and prostacyclin analogues. Upfront initial combination therapy in treatment-naive and newly diagnosed PAH patients improves symptoms, exercise capacity and outcome compared with initial monotherapy [1].

Despite the availability of multiple drugs interfering with the endothelin, nitric oxide and prostacyclin pathways, PAH is still characterized by high mortality rates. Considering the central role of dysregulated immune mechanisms in the pathogenesis of PAH, novel agents with anti-inflammatory and immunomodulatory properties could allow a step forward in the treatment approach. However, further investigations into safety and efficacy are needed prior to their approval.

In this review, we focused on the description of immune system abnormalities in patients with pulmonary arterial hypertension. The relevant, most recent research documents were identified in PubMed database. To obtain the most updated information on immunity pathways involved in pulmonary arterial hypertension, we acquired PubMed articles queried in February 2021. In our search, we used the following terms: “pulmonary arterial hypertension” AND “PAH”, “pulmonary hypertension” AND “immunotherapy”, “immune system”. Furthermore, the Library of the Cochrane Association was consulted for trials, as well as the Web of Science database, to identify the eligible studies. The analysis was restricted to abstracts published in English. The eligible studies for this review were selected based on full-length articles, both randomized controlled trials (RCT) and non-RCT, and non-controlled trials. The observational studies, data articles and brief and case reports were also included if they were sufficiently representative, and the information provided was meaningful. Conference abstracts and papers were excluded. Both human and animal studies were included. Pulmonary hypertension (PH) induced through hypoxia was found not eligible, since it is defined as PH classification group 3 and, thus, is not equivalent to PAH.

## 2. Immune System Cells

### 2.1. NK Cells and T Cells in Pulmonary Arterial Hypertension

NK cells (natural killer cells) are cells capable of identifying and eliminating cells infected with viruses or neoplastic lesions, and do so with the participation of FasL and TRAIL molecules and by degranulation of perforin-containing granules and granzymes [9,10]. The immune response towards antigens does not require the presence of MHC molecules, which allows reaching the maximum of its activity within 4–6 h after infection. Activating and inhibiting receptors tightly regulate the function of NK cells. This avoids attacks upon the body’s own cells [11,12]. Individual alleles of MHC class I molecules are recognized by the appropriate NK cell inhibitory receptors (the most common of which are CD94/NKG2A receptors), as well as some of the KIR family of receptors [11,12]. The lowering of the expression of MHC I molecules in tumor cells or virus-infected cells allows NK cell responses to be directed against them [13]. These cells then receive the appropriate signal from their own activating receptors. These receptors include NKp30, NKp44, NKp46, CD16 and some of the KIR superfamily receptors [14,15]. The expression of ligands for activating receptors can be induced on target cells or produced by other cells such as monocytes, macrophages and dendritic cells as a result of their contact with pathogens [11].

Recently, a growing body of data confirms the role of NK cells and CD8+ T cells in the regulation of vascular remodeling and regeneration [16]. Studies presented by Ormiston et al. [17] show that the NK cells of PAH patients produce an increased amount of matrix metalloproteinase 9, which has an effect on vascular remodeling and functional damage to NK cells. CD8 + cytotoxic T cells are a major component of the inflammatory response in the vascular plexus lesions of PAH patients [18]. Moreover, patients with pulmonary arterial hypertension and CD8+ T cell deficiency had a worse prognosis, and in experimental studies, rats deficient in T cells developed PAH more frequently [19].

Pulmonary arterial hypertension may be a complication of infections in which lymphocyte deficiency is observed [20]. In most of the accompanying disease states, PAH is associated with a defect in CD4 T cells and is interrelated with HIV or HHV-8 infection. This can be a complete CD4 cell deficiency, a decreased CD4/CD8 ratio or a decreased percentage of CD4+ CD25+ cells [21,22]. Taraseviciene-Stewart et al. [23] and Ormiston et al. [24] observed that experimental administration of NK cells and T cells prevented the development of pulmonary arterial hypertension. NK cells derived from patients with PAH also show an elevated production of matrix metalloproteinase-9. NK cell impairment leads to the increased production of IL-23 by pulmonary macrophages and other myeloid cell types. It may contribute to PAH pathogenesis by the abnormal regulation of pathological pulmonary vascular remodeling [25].

### 2.2. Regulatory T Cells in Pulmonary Arterial Hypertension

CD4+ T helper cells constitute a heterogeneous population of cells that play an important role in specific immunity [26]. Indeed, Treg cells with the CD4+/CD25+/Foxp3+ phenotype are crucial in maintaining immune system homeostasis, particularly in situations of long-lasting inflammatory response. They inhibit the activity of other T cells (especially CD8+ cytotoxic T cells), and limit immune response [27]. Regulatory T lymphocytes are divided into natural Treg lymphocytes (nTreg) which express CD25 and FOXP3, and induced (iTreg) or adoptive (aTreg). Natural Treg lymphocytes come from the thymus and constitute a total of about 5–10% of CD4+ T lymphocytes [28]. Natural Treg lymphocytes inhibit autoreactive T cells from autoimmune reactions. Adoptive T lymphocytes, in turn, acquire immunosuppressive functions upon contact with an antigen [29,30]. Treg CD4+ CD45RA+ FoxP3low lymphocytes are referred to as “resting lymphocytes”. CD4+ CD45RA− FoxP3high Tregs are activated Tregs and have strong immunosuppressive properties. CD4+ CD45RA− FoxP3low Tregs, however, do not have immunosuppressive properties, rather they produce the pro-inflammatory cytokines IL-2 and interferon gamma [31,32].

The maintenance of immune homeostasis and autotolerance, in which Treg is involved, is carried out by the ability of lymphocytes to secrete IL-10 and TGF-β1, hence, inhibiting the proliferation of effector T lymphocytes through the expression of the surface CD25 molecule and contact with antigen-presenting cells through surface molecules such as CTLA-4. These reduce the activation of effector T cells [33,34,35]. There are reports in the available literature on the protective role of regulatory T lymphocytes against pulmonary vascular endothelial damage in the course of pulmonary arterial hypertension, and that a disturbance in Treg function may increase the likelihood of developing PAH [36]. In addition, recent studies in animal models highlight that exposure to severe PAH development is increased as a result of Treg cell deficiency, especially in women [37].

### 2.3. B Lymphocytes in Pulmonary Arterial Hypertension

B cell differentiation is stimulated by CD4 + T helper cells. In turn, activated B lymphocytes produce antibodies, which may explain the presence of elevated levels of antinuclear antibodies found in patients with pulmonary hypertension [38]. The anti-endothelial autoantibodies promote the apoptosis of the endothelium. This contributes to the vascular remodeling that plays a crucial role in the PAH pathogenesis [39].

Studies by Ulrich et al. [40] have also shown that B lymphocytes of patients with confirmed PAH show increased RNA expression compared to healthy subjects, which may indicate that B lymphocytes of PAH patients are activated. Furthermore, in the lung tissue of patients with confirmed PAH, infiltration of pathological antibody-producing B lymphocytes has been demonstrated [41]. Breitling et al. [42] described the axis of mast cells—B lymphocytes involved in the remodeling of vessels and the development of pulmonary hypertension. The mediator connecting these cells is IL-6, which is synthesized and secreted by mast cells [43,44]. IL-6 has been shown to be a powerful prognostic factor in PAH patients, and increased IL-6 expression promotes the process of vascular remodeling [45,46,47]. Breitling et al. [42] also attempted to establish the role of B lymphocytes in the pathogenesis of PAH. According to the results of their experimentation, the administration of anti-CD20 antibodies or the genetic deficiency of B lymphocytes significantly improved the hemodynamics of the pulmonary circulation expressed by the reduction of RVSP and the reduction of vascular remodeling. In another experimental PAH model, Mizuno et al. [48] observed that depletion of B cells inhibited vascular remodeling.

Additional study has revealed that chronic activation of B lymphocytes leads to pulmonary fibrosis and increased autoimmunity in patients with systemic sclerosis [49,50]. Due to the large influence of the B-cell population on the progression of PAH associated with systemic sclerosis, a phase II clinical trial with the anti-CD20 antibody is currently being conducted [51].

The role of the immune system cells in PAH has been summarized in Figure 1.

### 2.4. Macrophages in Pulmonary Arterial Hypertension

Macrophages are key cells that coordinate the inflammatory response, but they have been shown to be associated with the pathogenesis of PAH [52,53]. These cells can be broadly classified into the subset of classically activated M1 (proinflammatory macrophages) and alternative activated M2 (anti-inflammatory macrophages). There are two types of tissue macrophages anatomically in the lungs: alveolar macrophages (AM) and interstitial macrophages. Inflammation can expand both macrophage populations through the proliferation of local macrophages and the recruitment of monocytes, which then differentiate into macrophages [53]. Regulation of the ratio of macrophages M1 and M2 must be conserved, as imbalance may have pathological consequences.

A study conducted by Vergardi et al. [52] showed that in the hypoxic-induced PAH mouse model, there is an early accumulation of macrophages in the lungs that acquire an activated M2 phenotype. Moreover, thanks to the use of transgenic mice with a specific induced expression of heme oxidase-1, an attenuation of macrophage accumulation and M2 activation was recorded, which was associated with the prevention of PAH [52]. The studies by Fan et al. [54] used the rat monocrotaline (MCT)-induced PAH model. Herein, lung tissue studies showed that CD68+ NOS2+ M1-like macrophages were significantly infiltrated early in the disease, but a significant decline was seen in the intermediate and late stages. In contrast, M2-like CD68+ CD206+ macrophages accumulated in lung tissue progressively from days 7 to 28. Additionally, it was observed that the relative ratio of M1/M2 macrophages increased over time. Fan et al. [54] also reported that M1 macrophages induced apoptosis of endothelial cells, and M2 macrophages significantly influenced the proliferation of endothelial cells and smooth muscle cells. In other studies, an increased number of CD68 + macrophages (differentiation cluster) has been reported in both the lungs of diseased patients and animal models of PAH [18,53].

Research by Zawia et al. [53] demonstrated that deletion of specific CD68+ macrophages in transgenic mice only led to the development of PAH in males, but not in females. Moreover, males were characterized by an imbalance in the M1/M2 ratio, confirming that this imbalance could affect the PAH phenotype. Additionally, a similar imbalance was observed in male bone marrow derived macrophages of transgenic mice, but also among monocyte derived macrophages of PAH patients after stimulation with doxycycline and IL-4 [53].

## 3. Cytokines

Many authors see a connection between excessive cytokine production and the development of PAH [45,55]. Soon et al., for example, confirmed [56] that in the plasma of patients with pulmonary arterial hypertension of various etiologies, a much higher concentration of various cytokines was found compared to controls. Their work has revealed that the concentration of IL-2, IL-6, IL-8 and IL-10 in the plasma influenced the survival of these patients, and the measurement of these parameters may be a useful biomarker to estimate the risk of developing PAH [46]. Furthermore, Cracowski et al. [56] conducted a study in a group of patients with hereditary pulmonary arterial hypertension that showed a statistically significantly higher risk of death in patients with significantly higher levels of IL-1α, IL-1β, IL-6, TNF-α and IL-13. The results that were gained were considered in the context of the age of the patients, the outcome of the 6-min walk test, the assessment of cardiac output and the pressure in the right atrium.

The attempt to answer the question about the relationship between the detected cytokine concentration and the clinical parameters of pulmonary hypertension assessment, however, demonstrates that increase in cytokine concentration is an independent factor associated with the mortality of patients [56]. Inflammatory cytokines and chemokines are responsible for the recruitment of immune cells, the activation and proliferation of pulmonary arterial smooth muscle cells and endothelial dysfunction. These features play an important role in PAH pathogenesis.

### 3.1. IL-2

The growth factor IL-2 is secreted mainly by antigen-activated CD4+ T cells and CD8+ T cells, activated dendritic cells, NK and NKT cells, as well as B cells [57]. Pleiotropic, autocrine and paracrine biological activity of IL-2 occurs after binding to specific receptors, which consist of three subunits: IL-2Rα (CD25), IL-2Rβ (CD122) and IL-2Rγc (CD132) [58]. IL-2 stimulates the proliferation and differentiation of T lymphocytes, causes the activation and proliferation of regulatory T lymphocytes and induces the differentiation of NK cells [59].

IL-2 has been found to play a significant role in suppressing the immune response, doing so most likely by influencing the development of CD4+ CD25+ regulatory T cells [60,61]. The inhibitory effects of IL-2 are based on the principle of negative feedback. Here, various mechanisms are at play. The first is the transient nature of IL-2 production. In the absence of antigenic stimulation, activated T cells die due to the lack of this cytokine in their microenvironment. The second mechanism is the initiation of the proapoptotic pathway by enhancing the expression of FasL on activated T lymphocytes, leading to their programmed death [62]. IL-2 also acts as a growth factor for NK cells, and stimulates these to secrete cytokines such as TNF-α, IFN-γ and GM-CSF. Additionally, it increases their cytotoxic activity [61].

There have been many clinical trials using IL-2 that have demonstrated the importance of this cytokine in the promotion of clonal expansion of regulatory T cells and immunomodulation of autoimmune diseases. The use of a low dose of recombinant IL-2 allowed induction of immune tolerance and proliferation of T lymphocytes, which results in the suppression of unwanted immune responses and a therapeutic effect in some autoimmune diseases. Zhu et al. showed [63] in PAH patients that IL-2 can enhance the suppressive effect of CD8+ CD25+ Foxp3+ T cells.

Increased levels of IL-2 and IL-1 have been observed in patients with chronic heart failure. In experimental animal studies, researchers found that IL-2 may increase vascular permeability, which causes pulmonary edema [46,64,65,66,67]. Ferro et al. [68] saw that IL-2 promotes vasoconstriction and pulmonary hypertension. Subsequent studies confirmed the participation of this cytokine in the pathogenesis of iPAH. Additional researches have also indicated that IL-2 increases the expression of endothelin, which accompanies the development of pulmonary hypertension [46,65].

### 3.2. IL-6

IL-6 stimulates the differentiation of B lymphocytes, monocytes and macrophages, and the activation of T lymphocytes. IL-6 production occurs in response to stimulation of macrophages, fibroblasts, dendritic cells (DC), lymphocytes and epithelial cells. IL-6 induces acute phase protein production by hepatocytes. In addition, it promotes B cell activation and monocyte differentiation. Research has indicated that IL-6 not only enhances pro-inflammatory effects, but also participates in anti-inflammatory responses. The anti-inflammatory effects of IL-6 include the ability to convert Th cells to Th2 by stimulating the production of IL-4 by Th2 cells. Another of its effects is the inhibition of IFN-γ production by Th1 lymphocytes [66]. It has been shown that the negative regulation of the inflammatory response by IL-6 is based on the induction of the suppressor of cytokine signaling-3 (SOCS-3) [67]. Rincon et al. and Chen-Kaminsky et al. have demonstrated that increased secretion of this cytokine accompanies many autoimmune diseases such as systemic lupus, rheumatoid arthritis, Castleman’s disease or juvenile idiopathic arthritis [69,70].

At the cellular and molecular level, vascular changes are associated with the activation of vascular endothelial growth factor (VEGF), mitogen-activated kinase (MAPK), increased transcription of the protooncogenic c-MYC/MAX complex, antiapoptotic survival, Bcl-2 and decreased growth inhibitor expression—specifically, TGF-β—transforming growth factor [71], as well as proapoptotic kinases—JNK and p38. These data indicate that IL-6 is involved in the development of distal proliferative arteriopathy, which in turn leads to an increase in pulmonary resistance and the development of pulmonary hypertension [47].

IL-6 was found to accompany the development of hypoxia-induced pulmonary hypertension and is an important factor in the pathogenesis of PAH [14]. A significantly elevated level of IL-6 in the serum and lungs of PAH patients has been demonstrated [45,47], which was associated with an unfavorable prognosis [46,56,72,73]. In a model of transgenic mice, IL-6 was noted to induce the pathological changes seen in advanced PAH, including muscular remodeling of distal pulmonary arterioles and convolutional arteriopathy, leading to increased pulmonary resistance (PVR) and PAH. However, despite the association of IL-6 with increased mortality of patients [46,72], Prins et al. [74] showed no correlation between plasma IL-6 concentration and mPAP or PVR indices.

The investigation conducted by this team provides the basis for the authors to suggest that IL-6 may be independently associated with PAH [74]. Similarly, Jasiewicz et al. [75] found no correlation between the BNP concentration assessment and the results of the 6-min walk test, with IL-6 concentration. The results of the study confirmed that the assessment of plasma IL-6 concentration, despite the obvious correlation with the risk of mortality in patients with PAH, is not an indicator of heart failure [75]. Still, studies with tocilizumab, a monoclonal antibody against the IL-6 receptor, evidenced an improvement in clinical and hemodynamic parameters in PAH patients with systemic connective tissue disease and Castleman’s disease [76,77,78]. Moreover, bosentan—an endothelin receptor antagonist that is used in clinical practice in the treatment of PAH, shows its anti-inflammatory effect by reducing the level of IL-6 and ICAM-1 in the blood, and this correlates with hemodynamic improvement [79].

### 3.3. IL-10

IL-10 is a suppressive cytokine that has anti-inflammatory and vasoprotective properties and is produced by Th2 cells, regulatory T cells and monocytes. During the active inflammatory process, IL-10 inhibits the production by macrophages and Th1 lymphocytes of various pro-inflammatory cytokines (including IL-6 and IL-8) [80,81]. In addition, exogenous IL-10 counteracts proliferative vasculopathy in vivo by inhibiting penetration by inflammatory cells [82]. Moreover, it prevents proliferation of smooth muscle cells [82,83] and the expression of chemokines [81,82,83,84]. Elevated levels of the anti-inflammatory cytokine IL-10 in the serum of PAH patients may indicate a compensatory mechanism that antagonizes the inflammatory response [46,81,84].

In experimental studies in rats, IL-10 expression protected against monocrotaline-induced pulmonary hypertension [81]. Attempts to use IL-10 in various autoimmune diseases (type 1 diabetes, multiple sclerosis, rheumatoid arthritis and psoriasis) have, however, produced divergent and inconsistent results. A high level of IL-10 in systemic lupus is considered pathogenic and its reduction has a positive effect on the course of the disease. Furthermore, IL-10 can promote a humoral immune response by increasing B cell expression and inducing antibody production [85].

### 3.4. IL-4

IL-4 is produced by Th2 lymphocytes, mast cells, NKT lymphocytes and basophils. It enhances the proliferation, activation and production of IgE and IgG1 antibodies by B lymphocytes, induces the proliferation of T lymphocytes and their differentiation towards Th2, as well as activates mast cells, eosinophils, monocytes and macrophages [86]. Yamaji-Kegan et al. [87,88,89] indicate that systemic administration of the recombinant HIMF protein (mitogenic hypoxia induced factor) causes pneumonia mediated by VEGF and the IL-4 dependent pathway.

In the early stage of inflammation, HIMF significantly increases VEGF expression and reduces VEGFR2 expression in the lungs, while causing the influx of macrophages. This process was completely inhibited in IL-4 deficient mice. This suggests that the inflammatory response to HIMF is IL-4 dependent and increases pulmonary inflammation. Beyond the aforementioned, IL-4 produced by cells of the immune system in response to systemic administration of HIMF was discovered to increase apoptosis of pulmonary endothelial cells. Soon et al. demonstrated a much higher concentration of this cytokine in the plasma of patients with iPAH [46]. IL-4 also has properties that stimulate the production of antibodies by B lymphocytes, which is important in the development and progression of PAH [23]. Soon et al. noted that apart from the significantly higher concentration of IL-4 in patients with iPAH, higher concentrations of IL-2, IL-10 and IFN-γ are observable, compared to the concentration of these cytokines in a control group [59].

### 3.5. Interferons

INF-γ is a pro-inflammatory cytokine that bring about macrophage and endothelial cells activity, activates APCs, enhances phagocytosis and increases the cytotoxicity of cytotoxic T lymphocytes and NK cells. IFN-γ, in also acting as an effector cytokine, is a key regulator of macrophage activation through the JAK/STAT signaling pathway [90], and increases the expression of adhesion molecules, the influx of neutrophils and macrophages to the site of inflammation, as well as the expression of MHC type I and II molecules. In addition, INF-γ contributes to the progression of systemic autoimmune diseases, in particular, systemic lupus [91]. There are many reports that clinically significant PAH can be induced by INF type I therapy [92,93,94]. Interferon α and β are listed as potential PAH-inducing toxins in the recommendations for the diagnosis and treatment of pulmonary hypertension of the European Society of Cardiology [7]. What is more, almost half of all patients (48%) receiving IFN-α therapy have a reduced pulmonary gas diffusion (DLCO) assessment of lung function by ≥15%, which may indicate undiagnosed pulmonary vascular pathology. Patients receiving IFN-α therapy for hepatitis C also have elevated levels of endothelin-1 (ET-1), a key mediator involved in the pathogenesis of PAH, and ET-1 can be induced by interferon in vascular smooth muscle [95].

## 4. Immunoregulatory Molecules

### 4.1. CTLA-4

CTLA-4 (cytotoxic T cell antigen 4) (CD152) is a receptor protein belonging to the immunoglobulin superfamily. It is expressed mainly on the surface of activated T lymphocytes, and on that of the B lymphocytes and dendritic cells as well [96]. The ligands for this receptor are CD80 and CD86 molecules that are evident mainly on antigen presenting cells. The CTLA-4 molecule is an important element of the negative regulation of the immune response, and it, when combined with a specific ligand, inhibits T lymphocytes [96]. This occurs through two main mechanisms—by affecting the ability of antigen presenting cells (APC) to stimulate T cells (extracellular mechanism) and by attenuating signals sent to T cells (intracellular mechanism) [97]. The physiological function of CTLA-4 is to inhibit T cell responses to self-antigens by controlling the activity of regulatory T cells [98]. Increased CTLA-4 expression, especially with regard to Treg and CD8 + lymphocytes, leads to a reduction in the quality of the immune response to cancer cells. In contrast, decreased CTLA-4 expression predisposes to the development of inflammatory processes [99].

Exposure to antigens causes activation markers to appear on the surface of lymphocytes. Early activation markers (CD69 and CD71) and late activation markers (CD25 and HLA-DR) can be notably distinguished [100]. T cell activation is a process that requires a signal associated with the MHC recognition by the T-cell receptor (TCR) on T lymphocytes, as well as a costimulatory signal that includes the attachment of molecules CD80 (B7-H1) and CD86 (B7-H2) on antigen-presenting cells (APC), with a CD28 molecule on the T lymphocyte. Sufficient CD28 and CD80/CD86 molecules lead to lymphocyte proliferation, increasing the survival capacity of lymphocytes and enhancing differentiation through the production of growth factors such as IL-2.

Another molecule with affinity for CD80/CD86 molecules is CTLA-4. In contrast to the CD28 molecule, association with CTLA-4 does not result in the appearance of cell stimulating signals, rather, the resulting competition for the CD80/CD86 binding site prevents the formation of costimulatory signals. The number of CTLA-4 molecules that compete with CD80/CD86 determines whether the lymphocyte will be activated or anergic. CTLA-4 has properties that inhibit the activation of lymphocytes, while it has been shown that it is constitutive on Treg lymphocytes and enables their suppressor functions [101]. The importance of the participation of the CTLA-4 molecule is indicated by the activation of lymphocytes towards self-antigens in mice deficient in CTLA-4. Cardiotoxicity has been found to be a side effect of the use of antibodies to block the pathway in which CTLA-4 is involved [102].

Our own studies have shown that in a group of patients with iPAH, there is a much higher percentage of CD4+ T cells with immune checkpoint proteins expression (CTLA-4 and programmed cell death receptor 1—PD-1) than in controls [103]. Similarly, a significantly higher percentage of these lymphocytes was detected in this patient group, compared to other PAH groups. A significant increase in the percentage of T lymphocytes with the CD8 + CTLA-4+ phenotype compared to controls was also demonstrated in patients with iPAH. In the work of Austin et al. [104], circulating CD8+ T cells were found to be lymphocytes with the CD45RA+ CCR7- phenotype, i.e., effector lymphocytes with cytolytic activity. Moreover, the number of naive lymphocytes with the CD45RA + CCR7- phenotype was significantly lower compared to their control group [104].

### 4.2. CD200 and CD200R

CD200 and CD200R are membrane glycoproteins belonging to the immunoglobulin-like proteins that perform mainly immunomodulatory functions. CD200 is expressed by neurons, vascular endothelial cells and T and B lymphocytes [105], while CD200R is expressed on cells of the myeloid lineage (monocytes and macrophages) and on T and B lymphocytes [106,107]. The interaction between CD200 and CD200R results in the activation of intracellular inhibitors, especially RasGAP, ending in inhibition of cell effector functions. Research indicates that CD200R activation stimulates T cell differentiation into Treg cells, enhances indolamine 2,3-dioxyganease (IDO) activation, modulates cytokine release and stimulates the synthesis of anti-inflammatory IL-10 and TGF-β [108].

Walker and Lue suggest that pro-inflammatory stimuli may bring about the secretion of cytokines or other factors that reduce CD200 expression [109], or, alternatively, result in the loss of cytokines or factors that increase or maintain CD200 expression levels. Studies in animal models have shown that a reduction in CD200 glycoprotein levels is associated with inflammation and aging, and that supplementation with recombinant, soluble CD200 protein reduces the inflammatory response [109]. Furthermore, experimental in vitro studies indicate that suppression of the inflammatory response associated with CD200 is proportional to the level of cellular expression of CD200R [110].

The role of CD200 in the spread of cancer cells and hypersensitivity reactions is also highlighted. These proteins are capable of generating immune tolerance, as well as regulating cell differentiation, cell adhesion and chemotaxis. These molecules play a role in the hemostasis of the immune system by inhibiting the inflammatory response to both external antigens (pathogens, allergens, etc.) and internal factors that trigger the activation of inflammatory cells (hypoxia, tissue damage, etc.) [111].

The CD200-CD200R interaction aids in modulating inflammatory response [112]. This pathway has been shown to contribute to the reduction of the excessive inflammatory response occurring during infection or neoplastic disease, and to the inhibition of the autoimmune response [113]. Cells expressing the CD200 molecule show an activation phenotype, while interaction with the CD200R molecule changes the Th1/Th2 lymphocyte relationship. In addition, the CD200-CD200R interaction increases the production of cytokines produced by Th2 lymphocytes [114]. Lymphocyte stimulation with IL-2 and phytohemagglutinin (PHA) causes much more CD200 molecules to appear on CD4 + T cells than on CD8+ T cells [115]. The involvement of the CD200-CD200R pathway in the reduction of excessive immune response has been demonstrated in various inflammatory diseases [116]. The relationship between CD200R expression and inflammatory diseases was explored, for example, by Gao et al., who showed a negative correlation between the expression of the CD200R molecule on macrophages and the concentration of C-reactive protein (CRP) in rheumatoid arthritis [117].

In animal models, the reduction of CD200 and CD2000R expression was found to be the cause of neurodegeneration [118]. Mice lacking the CD200 antigen revealed an increase in the inflammatory response and cytokine production [119]. Furthermore, sarcoidosis patients with decreased expression of CD200R on monocytes showed increased production of pro-inflammatory cytokines [120]. In contrast, pro-inflammatory cytokines present at the site of inflammation stimulate CD200 expression, which in turn suppresses an excessive inflammatory response [115,121].

CD200 mediated signaling was noted as being dependent on the level of CD200R expression. Cells with low or very low expression of CD200R exhibit only minimal inhibition capacity, which was confirmed by measuring the reduced IL-8 secretion. However, cells that showed moderate or high CD200R expression are characterized by properties that suppressed the immune response [114]. Mice lacking the CD200 gene show an increased number of activated monocytes, and the lack of CD200R was found to be associated with enhanced tumor necrosis factor alpha (TNF-α) production in response to stimulation with lipopolysaccharide (LPS), as well as a lack of the ability to inhibit proinflammatory cytokine production [122,123].

Although research has revealed that CD200 particles circulating in plasma affect the expansion of Treg lymphocytes, their role is not fully understood [124]. Studies, however, indicate that the percentage of Treg cells is reduced when the CD200-CD200R pathway is blocked by anti-CD200 antibodies [125]. Furthermore, a correlation has been demonstrated between plasma CD200 concentration (sCD200) and the severity of the disease in dermatitis. The soluble CD200 molecule correlates with the concentration of the cytokine IL-6, so it has been suggested that this form of CD200 may be a pro-inflammatory marker. Research also suggests that the sCD200 molecule may block the CD200-CD200R interaction and weaken the suppressive functions of this pathway [124]. Similarly, the use of antibodies that block the CD200-CD200R interaction were noted to increase inflammation and tissue damage, while treatment with CD200R agonists was seen to reduce inflammation and limit tissue destruction [126,127,128].

Deficiency of CD200 molecules and CD200R molecules are responsible for inappropriate control of the inflammatory response [123]. Research has uncovered the fact that IL-4 can increase the expression of CD200 and CD200R—as observed in a microglial inflammatory response model [112,129,130]. In addition, IL-4 deficiencies induce activated cells to insufficiently express CD200 and CD200R molecules. This has been demonstrated in Alzheimer’s disease research [123]. Similar relationships of negative correlation between CD200R mRNA and IL-4 were found in various forms of epilepsy in children [130].

Our own studies demonstrate that patients with CHD-PAH, CTEPH and iPAH show significantly higher percentages of T lymphocytes with the phenotype CD4 + CD200 and CD8+ CD200 in patients with CHD-PAH, CTEPH and iPAH, as compared to controls [131]. In addition, the CHD-PAH and iPAH groups displayed significantly higher percentages of lymphocytes within the CD19 + CD200 phenotype than in controls. Analysis of the percentage of lymphocytes expressing CD200R, however, reveal a significantly lower percentage of these lymphocytes in each PAH group, as compared to controls.

In the light of the results obtained by other researchers, our own research seems to confirm the supposition that high expression of the CD200 molecule may be the result of cell responses to chronic cytokine stimulation. This outcome suggests that an increase in CD200 expression is a compensatory mechanism that reduces exaggerated cytokine production [121]. Our own research leads to the notion that the CD200-CD200R pathway can counterbalance the effect of cytokine secretion, and indicates the existence of impaired immunosuppressive function of this pathway in PAH patients. The demonstrated deficiency of the CD200R antigen, accompanied by a decrease in the concentration of IL-4, may intimate a relationship between these two disorders.

### 4.3. PD1/PD-L1

PD-1 and its ligands were first described in 1992. Shortly thereafter, the PD1/PD-L1 pathway was recognized as an important immune checkpoint. PD-1, when activated, inhibits T cell proliferation and its effector functions such as cytotoxicity or cytokine release. This pathway is also responsible for the differentiation of CD4+ T cells and FOXP3 + regulatory T cells, thus controlling their inhibitory function [132] as well.

PD-1 is a receptor encoded by the PDCD1 gene and is expressed mainly on the surface of activated T lymphocytes, B lymphocytes, Treg and NK cells. The PD-1 receptor has two ligands: PD-L1 (B7-H1 and CD274) and PD-L2. PD1-L1 is a surface glycoprotein manifested on the cells of many tissues. This protein can be found, for instance, on the surface of dendritic cells, B and T lymphocytes, or cancer cells [133]. B7-H1 is present in almost all tissues of the body, and most likely its regulation under normal conditions depends on post-transcriptional processing. However, it is known that this protein can be expressed on various types of cancer cells [98].

Cancer cells have the ability to avoid the body’s immune response. This effect is connected, inter alia, with the increased expression of PD-1 receptor ligands on their surface. Under normal circumstances, the PD1/PD-L1 pathway plays an important role in hemostasis and the protection of the immune system [134]. For example, the percentage of Treg cells is reduced after PD-1 becomes attached to ligands on antigen presenting cells (APC) [135]. In contrast, in the case of cancer cells, the PD-L1 pathway may protect them from cytotoxic T lymphocytes by disrupting the immune system. This comes about by means of two mechanisms [134]. In the first mechanism, PD-L1 overexpression occurs in metastatic neoplastic cells, which prevents the activation of new cytotoxic T cells. In the second, with regard to dendritic cells, increased PD-L1 expression results in inactivation of cytotoxic T cells. In both cases, the interaction between the PD-1 molecule and its ligand inhibits the proliferation of T cells, lowers the production of cytokines and decreases the recognition of cancer cells [136]. Blockade of the PD-1/PD-L1 pathway is a promising therapeutic target as it results in an enhanced immune response against cancer [137,138].

Activation of lymphocytes causes the proliferation of PD-1 molecules on their surface. This comes about 48 h after the action of the stimulating factor. At the same time, the expression of the PD-L1 molecule is increased [139]. The relationship between PD-1 and PD-L1 is a regulatory mechanism that limits the inflammatory response in the body. The interaction of these molecules protects tissues from destruction. This effect may be the result of the immune system’s response to the constant presence of pathogens during chronic infections [140,141]. The functional importance of the PD-1/PD-L1 pathway, hence, includes inhibition of proliferation and cytokine production dependent on TCR receptor induction on lymphocytes [141].

Expression of PD-1 plays a key role in the antiviral response of CD8 + T cells [142]. Increased PD-L1 expression on antigen presenting cells (dendritic cells, macrophages and B lymphocytes) and interaction with the PD-1 molecule brings about a situation wherein CD8 + T lymphocytes become depleted lymphocytes and lose immunological control over the developing infection [141,142]. Loss of reactivity towards the pathogen subsequently leads to the development of chronic infection [142]. Experimental confirmation that the PD-1/PD-L1 pathway prevents viral clearance was obtained by Latchman et al. Their work revealed that the absence of PD-1 in mice improves the antiviral response. Virus elimination was also achieved in mice lacking PD-L1 [143]. After the pathogen is removed from the body, PD-1 expression is lowered and lymphocyte function is restored. Therefore, blocking the PD-1/PD-L1 interaction restores the immune control of the infection. Still, despite the increase in antiviral function, the complete absence of PD-L1 in experimentally infected mice resulted in their death. The presence of PD-L1 allowed the animals to survive during chronic infection, indicating that PD-L1 is necessary to prevent the tissue destruction that may occur during an overactive immune response due to infection [144].

PD-L1 is a key mediator of tissue tolerance [142]. PD-L1 expression has been demonstrated in many immunologically competent cells, but also in tissues [139]. Research demonstrates that expression of PD-L1 on the surface of pancreatic cells inhibits the response of autoreactive lymphocytes and protects against tissue damage. Herein, PD-L1 reduces the production of pro-inflammatory cytokines that follows activation of CD4+ T cells. What is more, attestation of the role of this molecule in the protection of tissues against the initiation and progression of autoimmune disease lies in fact that the lack of PD-L1 contributes to the development of diabetes [142]. More evidence of a protective role for the PD-1/PD-L1 pathway lies in the fact that neutrophil infiltration of cardiac tissue and progressive inflammation occurs in T cell PD-1 deficient mice. This study confirmed that the presence of the PD-1 molecule on T lymphocytes is necessary for the interaction with PD-L1 (which is expressed on muscle cells), and that the PD-1/PD-L1 pathway plays a protective role against tissue destruction [145].

Because the increase in PD-1 expression also occurs in inflammatory diseases, the question arises whether these cells may have a depletion phenotype [146]. Petrelli et al. [146], for instance, noted that the expression of the PD-L1 antigen on APC cells and PD-1 on CD8+ T cells was increased in the joint fluid of patients with rheumatoid arthritis. In contrast, CD8+ T lymphocytes showed the ability to produce cytokines and to resist the influence of regulatory T lymphocytes. Further research indicated that PD-1-related signaling results in a reduction in the activation of phosphoinositide 3-kinase (PI3K), followed by a reduction in Akt kinase activation. In contrast, in synovial fluid cells, Akt kinase hyperphosphorylation was found in CD8+ T cells expressing PD-1 [147]. Concurrently with the PD-1/PD-L1 pathway, soluble forms of PD-1 (sPD-1, PD-1 soluble) have been detected. These were found to be extant in the synovial fluid in high concentrations. The presence of sPD-1 may counteract PD-1 dependent suppression by blocking interaction with APC, which explains why effector cells cannot be suppressed at the site of inflammation [148]. Riley et al. suggest that PD-1 may be a marker of cells that, despite expressing inhibitory markers such as CTLA-4, are continuously activated due to an insufficient number of inhibitory signals [147].

During inflammation, research suggests that the glycolysis pathway is activated, secretion of proinflammatory cytokines is observed, and CD8+T cells exhibit cytotoxic properties. Such lymphocytes are not depleted as described in chronic inflammation caused by cancer or viral infection. Hence, PD-1 may not be a depletion marker and may characterize activation phase cells exposed to antigen contact. Thus, PD-1 may represent a functional adaptation to environmental conditions [114,149,150]. Another explanation for the presence of markers of depletion on the cell surface may be the assumption that among the PD-1 positive effector cells, there are also depleted cells [146].

The PD-1/PD-L1 interaction can function as an inhibitory or activation pathway. The contradictory findings regarding this pathway are explained by the possible involvement of additional proteins that act as agonists or antagonists into cell signaling inhibition. Furthermore, blocking CTLA-4 in the absence of PD-1 aggravates the symptoms of PD-1 deficiency, indicating a common role for these two molecules in maintaining tolerance [151]. Here, cytokines such as IFN-gamma, IL-4 and IL-10 increase PD-L1 expression in cells, and growth factors such as IL-2 can overcome the inhibitory effect of the PD-1/PD-L1 pathway [151,152]. Negative signals in the absence of the positive signals that the cell receives can be, therefore, be tolerated. In contrast, strong positive signals are converted into negative signals. The expression of PD-1 and PD-L1 does not exclude the ability of lymphocytes to proliferate and for normal effector functions to occur [151].

The PD-1/PD-L1 pathway may be the cause of the depletion of cells that have a protective effect in PAH [153]. Treg cells, for example, can be inhibited by PD-L1, the expression of which has been demonstrated on endothelial cells [37]. In an experimental study of pulmonary hypertension based on Treg lymphocyte deficiency, researchers revealed that the number of PD-L1 molecules was significantly reduced in the lung and right ventricular tissue of female rats with pulmonary hypertension. Why male rats expressed higher PD-L1 was not, however, elucidated. However, the female sex was found to develop pulmonary hypertension with a much more severe course than did male rats. The results of these studies suggest that in Treg lymphocytes, PD-L1 deficiency inhibits their suppressive functions, which leads to damage to the pulmonary bed and contributes to the development of PAH [37]. Similarly, the evaluation of the percentage of myeloid-derived suppressor cells (MDSC) indicated a significantly higher percentage of these cells expressing PD-L1 in PAH patients, compared to controls. A positive correlation of mPAP with expression of PD-L1 on MDSC was also demonstrated. Thus, an increase in PD-L1 expression on MDSC may contribute to their depletion and mediate the maintenance of inflammation in patients with PAH [153].

The PD-1/PD-L1 pathway promotes the inhibition of the activation and expansion of effector T cells and the promotion of Treg differentiation and function. As a consequence, the interaction of PD-1 with PD-L1 limits the tissue destruction that can occur in the uncontrolled response of T lymphocytes towards the infectious agent and prevents the development of autoimmune diseases [139,144]. Research indicates that the PD-1-PDL1 pathway suppresses the immune response, but may also be responsible for actually activating the immune response [139]. Accordingly, studies have revealed that the functional consequence of PD-1/PD-L1 interactions is dependent on the strong signals transmitted from TCR and CD28. An increase in TCR and CD28 activating signals may, therefore, subsequently bypass the inhibition step and leave the cell in the activation step [137]. The complexity of immune system interactions in the course of PAH has been summarized in Figure 2.

Our own studies have shown significantly higher percentages of lymphocytes expressing the marker of early activation (CD69) and late activation (CD25) in all patients with PAH. We found that patients with CHD-PAH, CTEPH and iPAH had a significantly higher percentage of T cells with the CD4+ PD-1+ phenotype than in the control group. We also saw a significantly higher percentage of CD4+ T cells expressing the PD-L1 molecule in these groups, compared to controls. Similarly, significantly higher percentages of T lymphocytes with the phenotype CD8+ PD-1 + and CD8+ PD-L1+ compared to controls were noted in these groups of patients. Moreover, patients in each group had a significantly higher percentage of B lymphocytes expressing PD-L1, compared to the study group.

The role of the PD-1/PD-L1 pathway is not, however, well understood in inflammation-related diseases. Our own studies have shown that in a group of patients with PAH, signs of lymphocyte activation, strong secretion of proinflammatory cytokines and an increased percentage of non-functional Treg lymphocytes, accompanied by an increase in the percentage of lymphocytes expressing immunosuppressive molecules, are observed simultaneously. The obtained results may suggest the existence of similar disorders that have been found by other researchers [149,150]. We can possibly conclude, hence, that in diseases accompanied by chronic inflammation, the expression of PD-1 and PD-L1 molecules does not indicate functional depletion of lymphocytes, but may characterize a population that is constantly exposed to stimulants or is a sign of functional adaptation to a chronically inflamed environment. However, it cannot be ruled out that among lymphocytes expressing immunosuppressive antigens, there are also those that become depleted by chronic stimulation. Yet another way to explain the presence of immunosuppressive molecules on the surface of lymphocytes (which in chronic viral diseases suggest silencing of the immune system), may be the functional exhaustion of cells that play a regulatory role in PAH.

## 5. Immunotherapeutic Approach

Certain research indicates that inflammation leads to the development of PAH. In animal models, the use of glucocorticoids and IL-1 receptor antagonists were found to prevent the development of PAH when applied at an early stage, in MCT-exposed rats. The potential anti-inflammatory glucocorticosteroid mediated mechanism is a reduction in IL-6. A direct anti-inflammatory effect on pulmonary artery smooth muscle cells has also been observed in an animal model. What is more, anti-inflammatory therapies were noted to play a role in human PAH (especially PAH associated with a strong inflammatory profile, e.g., CTD-PAH) by reducing pulmonary vascular remodeling, and, hence, bringing about an improve clinical outcome [154].

In PAH patients, tyrosine kinase inhibitors (TKIs) inhibit platelet-derived growth factor and are potent antiproliferative agents. The most studied TKI in PAH is imatinib. However, while it showed a decrease in PVR and mPAP in a randomized control phase III trial (IMPRES study), the functional NYHA status and time to clinical worsening did not improve. Additionally, treatment with imatinib caused more serious adverse events and a higher mortality rate than in a control group [155].

In contrast, in animal rodent models, anti-TNFα immunotherapy was reported to be effective in suppressing PAH progression and restoring BMP6/NOTCH2 signaling [156]. In other research, upregulation of TGF-β signaling was found in a BMPR-2 mutation rat model, and the TGF-β antagonist markedly ameliorated PAH [157].

Concerning the immune and the inflammatory component of PAH, targeted immunotherapy may be effective in preventing progression of the disease. One strategy is centered upon administration of elastase. This has been found to be effective in reversing advanced PH in an inflammatory monocrotaline model. The human recombinant elafin, a highly selective elastase inhibitor, has received Food and Drug Administration approval as a drug to treat PAH. Elafin was also found to suppress NFkB activation and the subsequent inflammatory response in an experimental model of PH [158].

Another therapeutic option is a low-dose of FK506 (tacrolimus), as this substance can reverse severe PH and the formation of neointima in the SUGEN/hypoxia rat model. In pulmonary artery endothelial cells derived from patients with idiopathic PAH, a low-dose of FK506 improved BMPR2 signaling. Moreover, in a single-center, phase IIa randomized clinical trial, a low-level therapy involving FK506 administration improved the BMPR-2 expression in peripheral blood mononuclear cells. Still, the serological and echocardiographic parameters of heart failure were not significant [159,160].

In an animal model, some effect on the restoration of BMPR-II signaling animal was shown by administration of berberine, puerarin and ataluren [161]. What is more, apelin expression was seen to decrease in PAH and the administration of apelin or BMPR-II ligands may improve it [162]. Apelin is an agonist of the apelinergic pathway, as well as a systemic vasodilator that is important in the homeostasis of pulmonary circulation.

Another potent signaling pathway that may be utilized in PAH pharmacotherapy is the inhibition of Rho-associated protein kinase (ROCK). The ROCK inhibitors, such as azaindole-1 and 18β-glycyrrhetinic acid (18β-GA) have been shown to be effective in animal studies. In these, azaindole-1 was noted to be capable of lowering pulmonary and arterial pressures and reversing hypoxic pulmonary vasoconstriction in rats [163], while 18β-GA has shown antioxidant, anti-inflammatory and anti-proliferative properties [164]. The most promising ROCK inhibitor is fasudil, which has revealed its effectiveness in a single-center clinical trial in 60 patients with severe PAH associated with congenital heart disease. The 60 mg dose of fasudil administered intravenously reduced significantly the pulmonary artery pressure without significant side effects [165].

## 6. Conclusions

A number of clinical studies have shown that disorders of the immune system play an important role in the development and maintenance of PAH. Registered dysregulation at the level of immune cells, the cytokines they release and, moreover, the negative effects noticed due to the action of immunoregulatory molecules indicate the complexity of this problem. Certainly, more detailed research on these elements in the context of the pathogenesis of PAH is required so that the results can be used to develop more effective diagnostics of the disease, but also to enable the development of targeted therapies that will allow patients to be cured.

The understanding of PAH immunity has led to the development of novel drugs affecting various mechanisms. Recently established drugs, such as fasudil or FK506 have shown promise in the treatment of PAH. The new insights into immunological, genetic, metabolic and endocrine disorders have greatly extended our knowledge and showed new roadmaps of potential therapeutic pathways. We believe that combination therapy, approaching different pathways is the future of PAH therapy. Despite the great progress in current pharmaco- and immunotherapy, based on safety and efficacy issues, new targeted agents should be developed to improve the prognosis of patients with PAH.

## Figures and Tables

**Figure 1 jcm-10-03757-f001:**
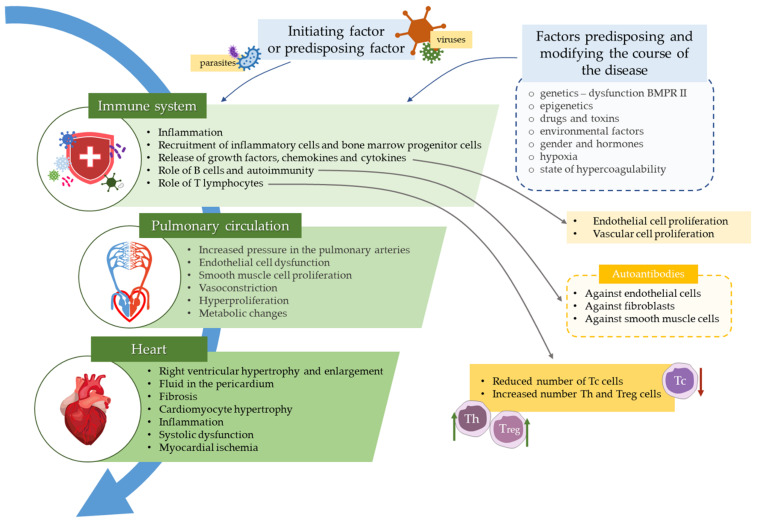
Contribution of the immune system cells and inflammation in the pulmonary arterial hypertension. Here, initiating or predisposing factors lead to immune system stimulation, which results in endothelial and vascular cells proliferation, autoantibodies formation, T helper cells (Th) and T regulatory cells (Treg) proliferation, as well as reduction of T cytotoxic cells (Tc). All these phenomena affect pulmonary circulation and result in heart damage.

**Figure 2 jcm-10-03757-f002:**
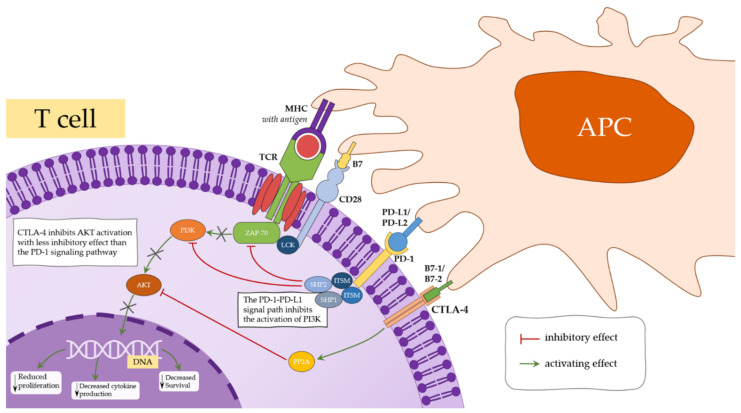
The main immunity pathways involved in PAH.

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
