# Peer review of "Role of the Immune System Elements in Pulmonary Arterial Hypertension"

_jcm, 2021, doi:10.3390/jcm10163757_

Round 1

Reviewer 1 Report

The authors provided an improved revised version of the manuscript compared to the previous one. However, the paper still suffers from some limitations:

  • Regarding the literature search, a) the authors should report the source used for the identification of relevant documents (Pubmed database) in the methodology section of the paper; b) PRIMA ckecklist has not been included in the manuscript;
  • The introduction is still too long, confusing and incorrect in some points: a) the description of some specific forms of pulmonary hypertension is notfunctional to the aim of the paper; b) the latest guidelines for the diagnosis and treatment of Pulmonary Hypertension date back to 2015 (line 85 of the paper – “According to the latest guidelines updated at the Sixth World Symposium on Pulmonary Hypertension 85 (WSPH) in 2018”).The new hemodynamic definition reassessed by the 6thWSPH Task Force is not yet included in ESC/ERS guidelines.
  • Pay attention to the bibliography, being references incorrectly numbered (e.g. first reference is n. 200);
  • As suggested, the authors added a figure representing the main immunity pathways involved in PAH but a short description of the figure is missing. It may help readers to focus on what is expressed extensively in the full text.
  • Concerning the english editing, I suggest you to have a native English speaker revisor to improve the quality of the paper making reading even more fluent.

Despite the improved version of the manuscript provided by the authors, considering the above limitations and concerns, the paper still needs a major revision.

Author Response

Dear Reviewer,

Thank you again for the consideration of our paper. Below we present answers your concerns.

 The authors provided an improved revised version of the manuscript compared to the previous one. However, the paper still suffers from some limitations:

  • Regarding the literature search, a) the authors should report the source used for the identification of relevant documents (Pubmed database) in the methodology section of the paper; b) PRIMA ckecklist has not been included in the manuscript

RE: The relevant, most recent research documents were identified in PubMed database. To obtain the most updated information on immunity pathways involved in pulmonary arterial hypertension we acquired PubMed articles queried in Febuary 2021. We used the following query term “pulmonary arterial hypertension” AND “PAH”, “pulmonary hypertension” AND “immunotherapy”, “immune system”. Furthermore, the Library of the Cochrane Association was consulted for trials, as well as the Web of Science database, to identify the eligible studies. The analysis was restricted to abstracts published in English.

The eligible studies for this review were selected based on full-length articles, both RCT and non-RCT, non-controlled trials. The observational studies, data articles and brief and case reports were also included if they were sufficiently representative and the information provided was meaningful. Conference abstracts and papers were excluded. Both human and animal studies were included. The pulmonary hypertension (PH) induced through hypoxia was found not eligible, since it is defined as another, PH classification group 3 and, thus, is not equivalent to PAH.

In our opinion, PRISMA Checklist concerns systematic reviews, and our paper was not dedicated to be one like that.

  • The introduction is still too long, confusing and incorrect in some points: a) the description of some specific forms of pulmonary hypertension is notfunctional to the aim of the paper; b) the latest guidelines for the diagnosis and treatment of Pulmonary Hypertension date back to 2015 (line 85 of the paper – “According to the latest guidelines updated at the Sixth World Symposium on Pulmonary Hypertension 85 (WSPH) in 2018”).The new hemodynamic definition reassessed by the 6thWSPH Task Force is not yet included in ESC/ERS guidelines.

RE: The introduction has been shortened and the misleading sentances were removed.

  • Pay attention to the bibliography, being references incorrectly numbered (e.g. first reference is n. 200)

RE: We have corrected the number of references, thank you.

  • As suggested, the authors added a figure representing the main immunity pathways involved in PAH but a short description of the figure is missing. It may help readers to focus on what is expressed extensively in the full text.

RE: The description of the figure has been added, thank you for this remark.

  • Concerning the english editing, I suggest you to have a native English speaker revisor to improve the quality of the paper making reading even more fluent.

RE: The paper has been corrected by the native speaker.

Once again, we would like to thank you for the time, effort, and consideration of our paper.

Kind regards,

Paulina Niedźwiedzka-Rystwej

Reviewer 2 Report

This review article discussed the regulated function of immune cells, released cytokines, and immunoregulatory molecules in the development of PAH. Immune-response in PAH is not an easy topic to dicuss.  Overall, I think this review article needs more description in depth. The review articles I would suggest the authors to read before submitting a revision are PMIDs:

  • 22215829
  • 24951765
  1. There is a minimum discussion on diagnosis and targeted immunotherapy that needs to be expanded.
  2. Most of the sessions overviewed the published data or results, can authors add their perspectives or comments?
  3. The "immune system cells" showed T cell, B cell, NH, what about macrophages, mast cells? Again, the majority of contents are to summarize other people's findings, not much of the author's views.
  4. No all the inflammable cytokines (such as CXCL12) were included. Can authors write more in-depth?

Author Response

Dear Reviewer,

Thank you again for the consideration of our paper. Below we present answers to your concerns.

This review article discussed the regulated function of immune cells, released cytokines, and immunoregulatory molecules in the development of PAH. Immune-response in PAH is not an easy topic to dicuss.  Overall, I think this review article needs more description in depth. The review articles I would suggest the authors to read before submitting a revision are PMIDs:

  • 22215829

  • 24951765

RE: The suggested papers have been added, thank you.

  1. There is a minimum discussion on diagnosis and targeted immunotherapy that needs to be expanded.

RE: The are several proof of evidence that inflammation play the role in the development of PAH. In animal models the use of glucocorticoids and IL-1 receptor antagonist prevented the development of PAH when used at early stage in MCT-exposed rats. The potential anti-inflammatory glucocorticosteroid mediated mechanism is a reduction in IL-6. The direct anti-inflammatory effect on pulmonary artery smooth muscle cells has also been observed in animal model. The anti-inflammatory therapies also play a role in human PAH (especially PAH associated with strong inflammatory profile, eg. CTD-PAH) by reducing the pulmonary vascular remodeling and through this improve clinical outcome [Price et al., 2012].

Tyrosine kinase inhibitors (TKIs) inhibit platelet-derived growth factor and are potent antiproliferative agents to use in PAH patients. The most studied TKI in PAH is imatinib. Although in the randomized control phase III trial (IMPRES study) it showed a decrease in the PVR and mPAP, the functional NYHA status and time to clinical worsening did not improve. Additionally the treatment with imatinib caused more serious adverse events and a higher mortality rate than in the control group [Hoeper et al, 2013].

The anti-TNFα immunotherapy was reported to be effective in suppressing PAH progression and restoring BMP6/NOTCH2 signaling in animal rodent models [Hurst et al. 2017].

The upregulation of TGF-β  signaling in BMPR-2 mutation rat models was found. The TGF-β antagonist markedly ameliorated PAH in this animal model [Tian et al, 2019].

Concerning the immune and the inflammatory component of PAH the targeted immunotherapy may be effective to prevent progression of the disease. One of the strategies is the elastase which has proved to be effective in reversing advanced PH in the inflammatory monocrotaline model. The human recombinant elafin, the highly selective elastase inhibitor has received Food and Drug Administration approval as a drug to treat PAH. Elafin also suppresses NFkB activation and the subsequent inflammatory response in the experimental model of PH [Rabinovitch et al. 2014].

The another therapeutic option is the low-dose of FK506 (tacrolimus) which can reverse severe PH and the formation of neointima in the SUGEN/hypoxia rat model.

In the pulmonary artery endothelial cells derived from patients with idiopathic PAH, the low-dose of FK506 improved the BMPR2 signaling. 

In a single-center, phase IIa randomized clinical  trial the low-level therapy of FK506 improved the BMPR-2 expression in peripheral blood mononuclear cells, however the serological and echocardiographic parameters of heart failure were not significant [Spiekerkoetter, 2017].

In animal model some the effect on the restoration of BMPR-II signaling animal was shown by berberine, puerarin or ataluren [Yang et al, 2020].

Apelin, the agonist of the apelinergic pathway, is as systemic vasodilator which is important  in the homeostasis of pulmonary circulation. The apelin expression is decreased in PAH and the administration of apelin or BMPR-II ligands may improve it [Yan J et. Al, 2020].

Another potent signaling pathway, which may be utylised in PAH pharmacotherapy is the inhibition of Rho-associated protein kinase (ROCK). The ROCK inhibitors, such as azaindole-1 and 18β-glycyrrhetinic acid(18β-GA) proved to be effective in animal studies. Azaindole-1 was capable of lowering pulmonary and arterial pressures and reversing hypoxic pulmonary vasoconstriction in rats [Pankey et al. 2012]. 18β-GA has shown antioxidant, anti-inflammatory and anti-proliferative in animal models [Zhang et al., 2019]. The most promising ROCK inhibitor is fasudil, which showed its effectiveness in a single-center clinical trial in 60 patients with severe PAH associated with congenital heart disease. The 60 mg dose of fasudil administered intravenously reduced significantly the pulmonary artery pressure without significant side effects [Ruan et al, 2019].

Refesences:

Price LC, Wort SJ, Perros F, Dorfmüller P, Huertas A, Montani D, Cohen-Kaminsky S, Humbert M. Inflammation in pulmonary arterial hypertension. Chest. 2012 Jan;141(1):210-221. doi: 10.1378/chest.11-0793. PMID: 22215829.

  1. Zhang, Z. Chang, P. Zhang, Z. Jing, L. Yan, J. Feng, Z. Hu, Q. Xu, W. Zhou, P. Ma, Y. Hao, R. Zhou

Protective effects of 18β-glycyrrhetinic acid on pulmonary arterial hypertension via regulation of Rho A/Rho kinsase pathway

Chem. Biol. Interact., 311 (2019), p. 108749

E.A. Pankey, R.J. Byun, W.B. Smith 2nd, M. Bhartiya, F.R. Bueno, A.M. Badejo, J.-P. Stasch, S.N. Murthy, B.D. Nossaman, P.J. Kadowitz

The Rho kinase inhibitor azaindole-1 has long-acting vasodilator activity in the pulmonary vascular bed of the intact chest rat

Can. J. Physiol. Pharmacol., 90 (7) (2012), pp. 825-835

  1. Ruan, Y. Zhang, R. Liu, X. Yang

The acute effects of 30 mg vs 60 mg of intravenous Fasudil on patients with congenital heart defects and severe pulmonary arterial hypertension

Congen. Heart Dis., 14 (4) (2019), pp. 645-650

Rabinovitch M, Guignabert C, Humbert M, Nicolls MR. Inflammation and immunity in the pathogenesis of pulmonary arterial hypertension. Circ Res. 2014 Jun 20;115(1):165-75. doi: 10.1161/CIRCRESAHA.113.301141. PMID: 24951765; PMCID: PMC4097142.

Spiekerkoetter E, Tian X, Cai J, et al. FK506 activates BMPR2, rescues endothelial dysfunction, and reverses pulmonary hypertension. J Clin Invest. 2013;123:3600–3613.

  1. Spiekerkoetter, Y.K. Sung, D. Sudheendra, V. Scott, P. Del Rosario, M. Bill, F. Haddad, J. Long-Boyle, H. Hedlin, R.T. Zamanian

Randomised placebo-controlled safety and tolerability trial of FK506 (tacrolimus) for pulmonary arterial hypertension

Eur. Respir. J., 50 (3) (2017), Article 1602449

Yang Y, Lin F, Xiao Z, Sun B, Wei Z, Liu B, Xue L, Xiong C. Investigational pharmacotherapy and immunotherapy of pulmonary arterial hypertension: An update. Biomed Pharmacother. 2020 Sep;129:110355. doi: 10.1016/j.biopha.2020.110355. Epub 2020 Jun 16. PMID: 32559622.

Yan J, Wang A, Cao J, Chen L. Apelin/APJ system: an emerging therapeutic target for respiratory diseases. Cell Mol Life Sci. 2020 Aug;77(15):2919-2930. doi: 10.1007/s00018-020-03461-7. Epub 2020 Mar 3. PMID: 32128601.

M.M. Hoeper, R.J. Barst, R.C. Bourge, J. Feldman, A.E. Frost, N. Galié, M.A. Gómez-Sánchez, F. Grimminger, E. Grünig, P.M. Hassoun, N.W. Morrell, A.J. Peacock, T. Satoh, G. Simonneau, V.F. Tapson, F. Torres, D. Lawrence, D.A. Quinn, H.-A. Ghofrani

Imatinib mesylate as add-on therapy for pulmonary arterial hypertension: results of the randomized IMPRES study

Circulation, 127 (10) (2013), pp. 1128-1138

L.A. Hurst, B.J. Dunmore, L. Long, A. Crosby, R. Al-Lamki, J. Deighton, M. Southwood, X. Yang, M.Z. Nikolic, B. Herrera, G.J. Inman, J.R. Bradley, A.A. Rana, P.D. Upton, N.W. Morrell

TNFα drives pulmonary arterial hypertension by suppressing the BMP type-II receptor and altering NOTCH signalling

Nat. Commun., 8 (2017), p. 14079
W. Tian, X. Jiang, Y.K. Sung, E. Shuffle, T.H. Wu, P.N. Kao, A.B. Tu, P. Dorfmüller, A. Cao, L. Wang, G. Peng, Y. Kim, P. Zhang, J. Chappell, S. Pasupneti, P. Dahms, P. Maguire, H. Chaib, R. Zamanian, M. Peters-Golden, M.P. Snyder, N.F. Voelkel, M. Humbert, M. Rabinovitch, M.R. Nicolls

Phenotypically silent bone morphogenetic protein receptor 2 mutations predispose rats to inflammation-induced pulmonary arterial hypertension by enhancing the risk for neointimal transformation

Circulation, 140 (17) (2019), pp. 1409-1425

  1. Most of the sessions overviewed the published data or results, can authors add their perspectives or comments?

RE: The understanding of PAH immunity led to the development of novel drug affecting various mechanisms. The recently established drugs, such as fasudil or FK506 are promising in the treatment of PAH. The new insights into immunological, genetic, metabolic and endocrine disorders have greatly extended our knowledge and showed new roadmaps of potential therapeutic pathways. In our belief the combination therapy, approaching different pathways is the future of PAH therapy. Despite the great progress in current pharmaco- and immunotherapy, based on the safety and efficacy issues, still new targeted agents should be developed to improve the prognosis of patients with PAH.

  1. The "immune system cells" showed T cell, B cell, NH, what about macrophages, mast cells? Again, the majority of contents are to summarize other people's findings, not much of the author's views.

RE: We have added a chapter on the matters (especially macrophages) as suggested by the Reviewer, thank you for your kind notice.

  1. No all the inflammable cytokines (such as CXCL12) were included. Can authors write more in-depth?

RE: We did our best to improve our comments in the paper on this subject.

Once again, we would like to thank you for the time, effort, and consideration of our paper.

Kind regards,

Paulina Niedźwiedzka-Rystwej